# Childhood Trauma and Dissociation Correlates in Alcohol Use Disorder: A Cross-Sectional Study in a Sample of 587 French Subjects Hospitalized in a Rehabilitation Center

**DOI:** 10.3390/brainsci12111483

**Published:** 2022-11-01

**Authors:** Grégoire Baudin, Servane Barrault, Hussein El Ayoubi, François Kazour, Nicolas Ballon, Damien Maugé, Coraline Hingray, Paul Brunault, Wissam El-Hage

**Affiliations:** 1Laboratoire de Psychopathologie et Processus de Santé, Université Paris Cité, F-92100 Boulogne Billancourt, France; 2QualiPsy, EE 1901, Université de Tours, F-37000 Tours, France; 3CHRU de Tours, Service d’Addictologie Universitaire, CSAPA-37, F-37000 Tours, France; 4Clinique Ronsard, Ramsay Santé, F-37170 Chambray-lès-Tours, France; 5CHRU de Tours, Clinique Psychiatrique Universitaire, F-37000 Tours, France; 6CHRU de Tours, Service d’Addictologie Universitaire, Equipe de Liaison et de Soins en Addictologie, F-37000 Tours, France; 7UMR 1253, iBrain, Université de Tours, Inserm, F-37000 Tours, France; 8Pôle Universitaire Adulte du Grand Nancy CPN, F-54520 Laxou, France

**Keywords:** childhood trauma, dissociative disorders, substance-related disorders, alcohol use disorders, mood disorders, anxiety disorders, posttraumatic stress disorder, dual diagnosis, addictive disorders, dual disorders

## Abstract

This study aimed to determine whether dissociative symptoms and childhood trauma (CT) may help identify a specific subgroup of patients among those hospitalized for alcohol use disorder (AUD). We assessed 587 patients hospitalized for an AUD in a French addiction rehabilitation center (cross-sectional study) regarding dissociative symptoms (DES-taxon), childhood trauma (CTQ), depression (BDI), anxiety (STAI-state and STAI-trait), posttraumatic stress disorder (PTSD; PCL-5), and AUD symptoms (AUDIT). We ran a hierarchical cluster analysis and compared the clusters in terms of dissociation and CT, as well as AUD, depressive, anxiety, and PTSD symptoms. We identified three clusters of patients: (1) patients with low AUD severity and low dissociation (LALD); (2) patients with high AUD severity and low dissociation (HALD); (3) patients with high AUD severity and high dissociation (HAHD). Patients from the HAHD group had significantly higher dissociation and more severe depression, anxiety, and PTSD symptoms than those with LALD and HALD. They also reported more emotional and sexual abuse than those with LALD. Among patients with an AUD, those with high dissociation may constitute an independent subgroup that exhibits a higher prevalence for CT and higher AUD severity, as well as higher depression, anxiety, and PTSD symptoms. Patients with more severe AUD and associated psychiatric symptoms should be systematically screened for dissociation and provided with tailor-based treatments.

## 1. Introduction

Childhood trauma (CT) is associated with severe mental health problems in adulthood [1], including depression [2,3], anxiety [4,5], posttraumatic stress disorder (PTSD, [4,6]), dissociative symptoms [7], and substance use disorders (particularly alcohol use disorders (AUD)) [8,9,10,11]. Lotzin et al. [12] found that 50.4% of subjects suffering from AUD reported at least one type of CT. In another study, Grundmann et al. [13] found that 36.2% of the participants with AUD have experienced either physical or sexual abuse. Lown et al. (2011) [14] found that specific types of CT, mainly physical and sexual abuse, are more prevalent in women suffering from AUD.

Widom et al. [15] reported indirect paths from child maltreatment to AUDs, suggesting the existence of mediating/moderating factors. Indeed, children exposed to CT are more likely to develop depressive, anxiety, or PTSD symptoms in adulthood. Furthermore, a fraction of patients with AUD exposed to CT is more likely to have comorbid PTSD, anxiety [16], or depressive disorder than children unexposed to CT [9]. Advocates of the self-medication hypothesis suggest that alcohol consumption could be a short-term strategy to cope with painful emotions, memories, or situations [17,18], as well as with PTSD symptoms [19].

As for dissociation, it has been extensively linked to exposure to CT [7] and, to a lesser extent and less consistently, to AUD [20]. Among patients with AUD, a history of CT has been associated with dissociative symptoms and/or disorders [21]. Schäfer et al. [22] found that dissociation, but not CT, is associated with the severity of AUD. One study showed that dissociative tendencies regulate relations between CT and AUD [23].

The authors supporting the chemical dissociation hypothesis [24,25] suggest that some subjects may use psychoactive substances, including alcohol, to achieve a chemically induced dissociative state to regulate painful emotions or memories and avoid psychological distress [23]. This hypothesis also implies that chemical and psychological dissociations are mutually exclusive. A study by Hingray et al. [26] investigated the relationship between AUD and dissociation, showing results that are in line with the chemical dissociation hypothesis. However, we could not find any study that tested which of these hypotheses better explains the relationship between CT, dissociation, and AUD. We do not know if the mutual exclusion assumption between psychological and chemical dissociation is valid either.

This study aimed to determine, among patients hospitalized for an AUD, whether dissociative symptoms may help identify a specific subgroup of patients with specific psychopathological characteristics. We hypothesized that among AUD patients, those with higher dissociation may represent an independent and more homogeneous subgroup and that these patients may present more severe psychopathology (i.e., CT exposure, as well as depressive, anxiety, and PTSD symptoms). We also aimed to identify whether other psychiatric symptoms, such as AUD severity, depression, anxiety, or PTSD symptoms, may be helpful in identifying more homogeneous subgroups.

## 2. Materials and Methods

### 2.1. Setting and Participants

This cross-sectional study was conducted at Le Courbat rehabilitation center, which treats AUD patients from all over France. This center was initially reserved for police officers from all over France suffering from burnout or addictive disorders. For more than ten years, this center has been admitting civilian patients living in the Centre-Val-de-Loire region of France.

All consecutive patients hospitalized for an AUD between January 2016 and October 2017 were recruited after giving written consent. AUDs were clinically determined in accordance with the International Classification of Diseases, Tenth Revision [27]. Eligible patients then completed self-administered questionnaires two weeks after detoxification using digital tablets or computers specifically provided for this study. The self-administered questionnaires were designed and filled in online using the Sphinx Mobile iQ 2 software. In total, we recruited 568 participants (87 women, 15.3%), with a mean age of 44.5 (SD = 9.0).

### 2.2. Measures

#### 2.2.1. Sociodemographic Data

Data about gender, age, education, and marital status were collected through self-administered questionnaires.

#### 2.2.2. Childhood Trauma

We used the French version of the Childhood Trauma Questionnaire (CTQ) short form [28] to assess CT. This 25-item self-rating scale retrospectively assesses five types of childhood maltreatment, namely childhood physical abuse (CPA), emotional abuse (CEA), sexual abuse (CSA), physical neglect (CPN), and emotional neglect (CEN). All items of this questionnaire begin with the phrase “When I was growing up, …”. Each of these types of CT is assessed with five items rated on a five-point Likert scale. Scores range from 5 to 25, with higher scores suggesting more severe CT. The French version of the CTQ has acceptable to excellent internal consistency (α ranging from 0.68 to 0.91) and test–retest reliability (r ranging from 0.73 to 0.94) [29].

#### 2.2.3. Dissociative Experiences

We used the Dissociative Experiences Scale-taxon (DES) to measure levels of dissociation among the participants. This self-rating scale includes eight items from the DES ([30]; French validation by Darves-Bornoz et al. [31]) and provides a sensible measure of pathological dissociation [32]. The DES-taxon requires participants to estimate the percentage of daily time they experience what the items describe. We used Waller and Ross’s [33] method to calculate the Bayesian probability of participants belonging to the DES-taxon. A probability higher than 0.90 suggests that the respondent belongs to the DES-taxon and is likely to match the criteria for a dissociative disorder [32].

#### 2.2.4. Alcohol Use Disorder

The intensity of AUD was assessed using the French version of the Alcohol Use Disorder Identification test (AUDIT) [34,35]. Scores range from 0 to 40, with higher scores suggesting more deleterious use of alcohol. The AUDIT has excellent internal consistency (α = 0.87) [35].

#### 2.2.5. Depressive Symptoms

The intensity of depressive symptoms was assessed using the shortened Beck Depression Inventory (BDI) [36], a 13-item self-rating scale. The total score ranges from 0 to 39, with higher scores indicating greater depression. The French validation [37] showed a positive and significant correlation (r = 0.41, *p* < 0.001) between BDI and Hamilton Rating Scale for Depression scores.

#### 2.2.6. Anxiety Symptoms

Anxiety severity was assessed using the French version of the State–Trait Anxiety Inventory (STAI) forms A and B [38,39], two 20-item self-rating scales that assess state anxiety (STAI-state) and trait anxiety (STAI-trait), respectively. The items are rated on a four-point Likert scale ranging from 1 to 4, with higher scores indicating higher anxiety. Scores range from 20 to 80. Internal consistency was excellent for state anxiety (α = 0.90) and trait anxiety (α = 0.91) [39].

#### 2.2.7. Posttraumatic Symptoms

We assessed PTSD symptoms using the PTSD checklist (PCL-5) [40,41], a 20-item self-rating scale. It assesses PTSD symptoms as described in the DSM-5 [42] using a Likert-type scale (from 0: not at all to 4: extremely). Items can be divided into four subscales: re-experiencing (DSM-5 criterion B, score of 0–20), avoidance (criterion C, score of 0–8), negative alterations in cognition and mood (criterion D, score of 0–24), and arousal (criterion E, score of 0–28). We used these sub-scores as measures of the severity of PTSD symptoms. The four sub-scores of the French version of the PCL-5 displayed acceptable to good internal consistency (α ranging from 0.79 to 0.87) and acceptable to excellent test–retest reliability (intraclass correlations (ICC) for criteria B, D, and E ranging from 0.78 to 0.92, with ICC = 0.66 for criterion C) [41].

### 2.3. Statistical Analyses

Data were analyzed using R version 3.6.1 [43]. In order to define homogeneous groups in terms of DES-taxon belonging and AUD severity, we conducted a cluster analysis. DES-taxon belonging was considered a binary variable (i.e., 0: absent or 1: absent), and the total AUDIT score was considered a numerical variable. Classical clustering methods are not suitable for mixed data (i.e., both binary and numerical variables). Thus, as Kassambara [44] recommended, we conducted hierarchical clustering on principal components (HCPC), a more appropriate clustering method for mixed data. Preliminary analyses revealed variable rates of missing data (e.g., the lowest rate of missing data was 0% for age and total AUDIT score, while the highest rate of missing data was 29.40% for DES-taxon belonging). In order to address this issue, we conducted multiple imputations with the missMDA R package [45].

HCPC first requires running a factor analysis for mixed data (FAMD). Thus, FAMD was conducted with the imputed data. Then, FAMD results were used to run HCPC. The optimal number of clusters was based on the inertia gain graphic. These analyses were performed with the factoMineR R package [46]. Following this, we compared the identified clusters regarding CT and clinical variables with ANOVAs and a Tukey’s posthoc test. For all models, our dependent variables (i.e., the CTQ, BDI, STAI-state, STAI-trait, and PCL-5 scores) were quantitative variables. In all of the ANOVAs, our only independent variable was the identified clusters. For the second run, we ran ANCOVAs in which we added covariates. The potential covariates were age (quantitative variable), sex (binary variable), education (binary variable: high school diploma/no high school diploma), and marital status (binary variable: in couple/single). All potential covariates associated with our dependent variables at *p* < 0.20 (determined using a t-test for binary variables and correlations for quantitative variables) were included in the ANCOVAs. Again, when the “cluster” variable appeared to be a significant predictor, we ran Tukey’s post hoc tests.

### 2.4. Ethics

An institutional review board approved this study prior to its beginning (CERNI Tours—Poitiers; July 2015). All of the data collected were in line with French recommendations regarding the use of personal data, with the approval of the French Commission Nationale de l’Informatique et des Libertés (CNIL).

## 3. Results

### Characteristics of the Participants

A total of 568 participants (87 women, 15.3%) were included. The mean age was 44.5 (SD = 9.0). The characteristics of the participants regarding education and marital status are displayed in Table 1.

Some of our variables of interest had missing data. Matrimonial status and Highest diploma had 1.58% and 7.04% of missing data, respectively. CTQ scores had a rate of missing data ranging from 1.58% (childhood emotional neglect) to 3.87% (childhood emotional abuse). Childhood physical abuse, childhood sexual abuse, and childhood physical neglect had missing data rates of 2.29%, 2.82%, and 1.76%, respectively. DES-taxon belonging had a missing data rate of 29.40%. Age, gender, and AUDIT score had no missing data.

The ANCOVAs among clusters showed specific differences for emotional and sexual abuse and physical neglect. Tukey’s post hoc analyses for the ANCOVAs revealed that those with HAHD had higher sexual abuse scores than those with LALD (*p* = 0.004) but not those with HALD (*p* = 0.065). Those with LALD and HALD did not have significantly different sexual abuse scores (*p* = 0.071). We observed the same pattern with regard to the emotional abuse scores (see Figure 1). However, all three clusters had significantly different physical neglect scores (all *p* < 0.05) (results are detailed in Table 2 and Figure 1).

#### Clinical Variables

We discovered a pattern similar to the one observed between the clusters regarding CT scores. Those with HAHD had the highest mean scores to clinical scores, followed by those with HALD and then those with LALD. The AUDIT scores were the only exception: Those with HALD had the highest scores. After adjustment on the pertinent covariates, the AUDIT scores significantly differed between the clusters (F = 674.67 (2,561), *p* <0.001). Tukey’s post hoc comparisons are available in Table 2.

The three clusters significantly differed regarding all of the clinical variables (all *p* < 0.001, see Table 2). Tukey’s post hoc comparisons for the ANCOVAs revealed no significant differences between those with LALD and HALD regarding BDI, STAI-state, and the five PCL-5 scores. However, we observed a significant difference between those with LALD and HALD for STAI-trait scores (*p* = 0.018, see Figure 2). Furthermore, for all of the clinical variables, those with HAHD significantly differed from the other two clusters (all *p* < 0.005).

For three out of four of the PCL-5 scores, those with HAHD had significantly different scores (all *p* < 0.001, except for PCL-5 criterion E) from the other two clusters. Furthermore, those with HALD and LALD had similar scores (see Figure 3).

## 4. Discussion

This study found that among patients hospitalized for AUD, those with high dissociation may constitute an independent and more homogeneous subgroup of patients. This subgroup is characterized by a more severe AUD, higher scores for depression, anxiety, and PTSD symptoms, as well as a higher prevalence for CT.

We initially hypothesized that we would distinguish four subgroups of patients: one in which participants would only and heavily use alcohol to regulate their painful memories and emotions (i.e., exogenous dissociators); a second in which participants would only use dissociation (i.e., endogenous dissociators); a third in which participants would use both endogenous and exogenous dissociation; and a fourth in which participants, while facing significant emotional distress, would not use these strategies to cope. We did not find a group of endogenous dissociators, which is unsurprising given the context of recruitment. Le Courbat is a rehabilitation center that primarily provides care to patients with addiction. Patients admitted in such settings preferentially use alcohol rather than endogenous strategies such as dissociation to cope with painful emotions. Furthermore, patients who preferentially use psychological dissociation to cope with painful memories and emotions (e.g., patients with dissociative disorders) are more likely to be diagnosed by and seek help from psychotherapists in outpatient settings. This selection bias may explain the absence of a group of endogenous dissociators in our sample. The presence of participants who did not belong to the DES-taxon and who did not have a high AUDIT score was more surprising. Given that we collected data at admission, these participants may represent a fraction of patients who still have sufficient psychological resources to cope with stressful events and who are still able to seek help to address their alcohol misuse issues.

Finally, the group comprising participants with both severe problematic alcohol consumption and dissociative symptoms leads us to question the chemical dissociation hypothesis. Furthermore, it may contradict the assumptions of the chemical dissociation hypothesis, which hypothesize that heavy alcohol consumption occurs when dissociative abilities are depleted [25].

Exposure to CT results showed the following pattern: for the five types of CT, those with LALD always had the lowest mean score, followed by those with HALD, while those with HAHD always had the highest CTQ scores. Regarding these results, we found only significant differences among the clusters for CEA, CSA, and CPN; all of the other types of CT were of equivalent intensity across clusters. We could have expected significant differences among the clusters for at least CPA [14]. For this type of CT, we cannot rule out a lack of statistical power that could have prevented us from finding differences in other types of CT.

Sexual abuse in childhood (CSA) is considered to be an adverse experience with high traumatic potential. In our study, participants with the lowest mean score were those who reported the lowest alcohol misuse problems and did not have high dissociative tendencies. The other two groups, comprising participants who reported serious alcohol misuse problems and, for a small proportion, high dissociative tendencies, reported more intense CSA, as found in previous studies [9,14,15]. CEA and CPN experiences are also distinct in the three groups. Repeated traumatic experiences of emotional abuse and physical neglect in childhood may lead to a feeling of insignificance and unworthiness regarding the persons who were supposed to take care of and love them. The addition of intense potentially traumatic experiences (i.e., CSA, CEA, CPN, and possibly CPA) may increase the risk that patients who already tend to seek relief with alcohol also have high dissociative tendencies. Therefore, a dose–response relationship between CT exposure and AUD severity is possible [1,47].

Our results highlighted significant differences among the groups regarding depressive, anxiety, and PTSD symptoms. First, subjects in the LALD subgroup significantly differed from those in the HALD subgroup regarding trait anxiety. These results support the hypothesis of a subgroup of patients who, despite their CT being globally similar to those who have problematic alcohol consumption, preserved psychological resources to cope with difficult memories or emotions in their daily life. Second, we measured depressive, anxiety, and PTSD symptoms two weeks after alcohol abstinence. This is in line with previous results showing that these symptoms decrease after a one-to-three-week interval after admission to a detoxification center [48,49]. Subjects in the LALD subgroup may particularly benefit from admission, at least regarding their trait anxiety. However, these patients have comparable levels of depressive, state anxiety, and PTSD symptoms to those in the HALD subgroup. Thus, patients in the LALD subgroup may suffer from less severe depression and anxiety before their withdrawal.

On another level, some patients who suffer from anxiety or mood disorders may preferentially self-medicate their anxiety/depressive symptoms with alcohol [50,51]. This short-term coping strategy has noticeable disadvantages since it increases the risk of persistent AUD [50,51]. The differences in the severity of depressive symptoms and trait anxiety between these two subgroups may stem from the significant differences in the severity of CT.

Subjects in the HAHD subgroup clearly differed from those in the two other clusters regarding depressive, anxiety, and PTSD symptoms. Childhood traumas experienced by these patients were more varied and severe compared to others, which is in line with previous results [52]. These early and repeated adverse experiences can explain the higher severity of psychiatric symptoms in patients with problematic alcohol consumption (Rehan et al., 2017) [11].

Our results suggest that patients with dissociative symptoms and cumulative traumatic experiences will suffer from severe depressive, anxiety, and PTSD symptoms that cannot be relieved by alcohol consumption. Alternatively, severe PTSD symptoms may extend beyond coping capacities, even after alcohol use. Thus, when heavy alcohol consumption is insufficient, patients tend to dissociate in order to avoid emotional distress. This hypothesis suggests that dissociation is a backup strategy to alcohol consumption and, thus, supports the self-medication hypothesis [19]. In our study, subjects were recruited while being hospitalized in a rehabilitation center, a least two weeks after their last alcohol consumption. While cessation of alcohol is generally associated with an improvement in depressive and anxiety symptoms [53], this specific subgroup of patients may not have benefited from it.

In addition, we can discuss that some patients may have a severe alcohol consumption pattern in order to alleviate painful symptoms without dissociation. One could also conclude that dissociation may hinder the relieving effects of alcohol. This hypothesis remains to be tested. Although our results support the self-medication hypothesis, we cannot reject the chemical dissociation hypothesis. Our sample represents only a fraction of persons with AUD. In different settings, it would have been possible to find a group of patients with both AUD and severe dissociative symptoms but with less severe depressive, anxiety, and PTSD symptoms compared to those with AUD only. It is also possible that there has been a shift in the requested emotional regulation strategies according to the severity of symptoms and/or over time. Roesler and Dafler [24] distinguished what they called “true addiction” from situational users or “chemical dissociators”. We hypothesized that situational users might develop real alcohol dependence over time through repeated heavy drinking, especially when they are facing severe depression or anxiety [54].

Our study has several limitations that need to be acknowledged. First, the cross-sectional design of our study has led us to find associations between traumatic experiences, clinical presentation, and substance use, but we cannot make formal conclusions about the causal relationship between these different factors. Therefore, our conclusions remain hypothetical on this point. Second, we collected data using self-reports, a time-saving but biased method. Regarding CT, there are concerns about recall bias, especially for subtle and repeated traumatic experiences, such as emotional abuse [55]. Furthermore, CTQ does not provide specific information about the age of exposure to CT. Taking into account such information would probably lead to a better understanding of the association between CT, alcohol use disorder, dissociation, and psychiatric symptoms. The DES-taxon, BDI, STAI-state, STAI-trait, and PCL-5 are also subject to recall bias and social desirability. Furthermore, one of our variables, DES-taxon belonging, had high rates of missing data. Although we alleviated this issue with multiple imputations, this is still a limitation in the interpretation of the results. Finally, recruitment took place in a rehabilitation center. Patients recruited in this facility represent a fraction of subjects suffering from AUD [56]. They also completed our questionnaires after two weeks of abstinence. Furthermore, only a fraction of patients who experienced CT and later developed AUD also had comorbid depression, anxiety, or PTSD [9,16]. Thus, our results must be interpreted with caution since they cannot be generalized to the whole population of subjects with AUD.

## 5. Conclusions

Our study showed the association between childhood trauma, dissociation, and depressive, anxiety, and PTSD symptoms. Although we highlighted the importance of severe dissociative symptoms in the overall severity of other psychiatric symptoms, we could not draw firm conclusions about causality. In order to confirm the hypotheses drawn from our results, prospective longitudinal studies. Such research projects should observe the temporality of the occurrence of various psychiatric symptoms. Such studies could help to differentiate patients with AUD more accurately according to their past exposure to CT and their dissociative symptoms.

Our results encourage the routine assessment of CT, and dissociative symptoms in patients admitted to rehabilitation centers for AUD. When mental health professionals identify patients with problematic alcohol consumption and dissociative symptoms, they should consider trauma-informed psychotherapy in addition to the usual treatments offered to these patients [57], as well as less harmful strategies to cope with painful emotions and memories.

## Figures and Tables

**Figure 1 brainsci-12-01483-f001:**
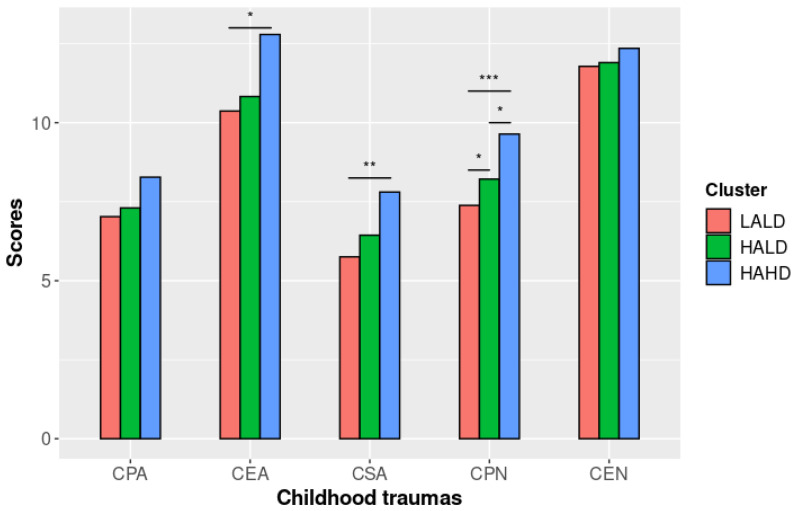
CTQ scores by cluster (adjusted models). Note. **LALD**: “Low Alcohol Low Dissociation” subgroup; **HALD**: “High Alcohol Low Dissociation” subgroup; **HAHD**: “High Alcohol High Dissociation” subgroup; **CPA**: Childhood physical abuse; **CEA**: Childhood emotional abuse; **CSA**: Childhood sexual abuse; **CPN**: Childhood physical neglect; **CEN**: Childhood emotional neglect; *: *p* < 0.05; **: *p* < 0.01; ***: *p* < 0.001.

**Figure 2 brainsci-12-01483-f002:**
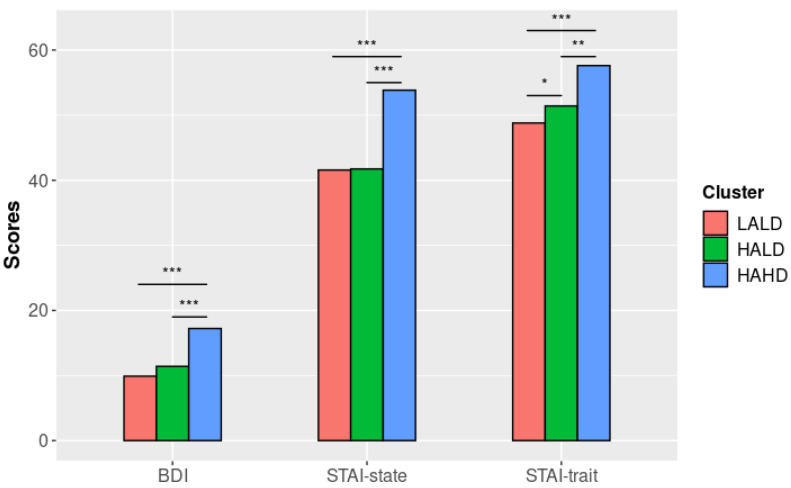
BDI, STAI-A, and STAI-B scores by cluster (adjusted models). **Note. LALD**: “Low Alcohol Low Dissociation” subgroup; **HALD**: “High Alcohol Low Dissociation” subgroup; **HAHD**: “High Alcohol High Dissociation” subgroup; **BDI**: Beck Depression Inventory; **STAI**: State–Trait Anxiety Inventory; *: *p* < 0.05; **: *p* < 0.01; ***: *p* < 0.001.

**Figure 3 brainsci-12-01483-f003:**
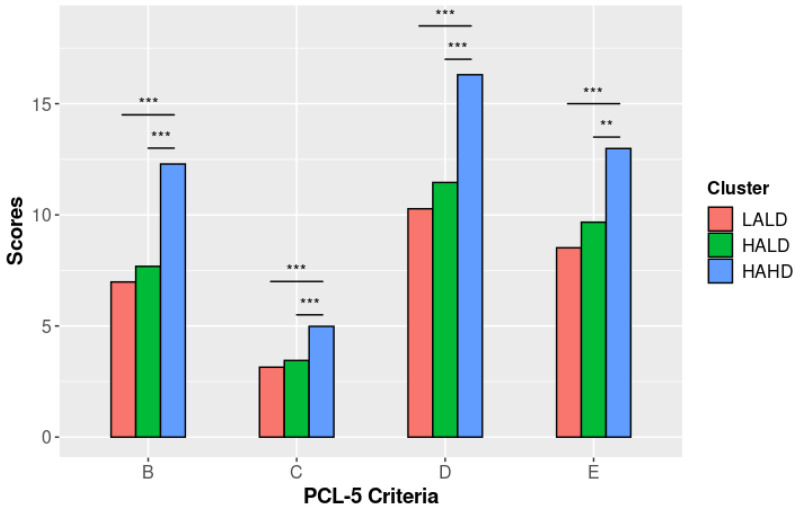
PCL-5 scores by cluster (adjusted models). Note. LALD: “Low Alcohol Low Dissociation” subgroup; HALD: “High Alcohol Low Dissociation” subgroup; HAHD: “High Alcohol High Dissociation” subgroup; **: *p* < 0.01; ***: *p* < 0.001.

**Table 1 brainsci-12-01483-t001:** Sociodemographic and clinical characteristics of the participants by cluster.

Variables	LALD (*n* = 160)	HALD (*n* = 372)	HAHD (*n* = 36)	Total
	Mean (SD)/*n* (%)	Mean (SD)/*n* (%)	Mean (SD)/*n* (%)	Mean (SD)/*n* (%)
Gender (women)	29 (18.13%)	52 (13.98%)	6 (16.67%)	87 (15.31%)
Age	45.8 (9.35)	44.1 (8.87%)	42.9 (7.83)	44.51 (8.97)
**Marital status**				
Single	45 (28.13%)	121 (33.24%)	14 (40.00%)	180 (31.69%)
In a relationship	65 (40.63%)	123 (33.79%)	14 (40.00%)	202 (35.56%)
Widowed, divorced, separated	50 (31.25%)	120 (32.98%)	7 (20.00%)	177 (31.16%)
**Highest diploma**				
Certificate of general education	18 (11.76%)	63 (18.42%)	8 (24.24%)	89 (15.67%)
Certificate of technical education	48 (31.37%)	114 (33.33%)	14 (42.42%)	176 (30.99%)
High school diploma	56 (36.60%)	90 (26.32%)	7 (21.21%)	153 (26.94%)
Undergraduate	15 (9.80%)	39 (11.40%)	3 (9.09%)	57 (10.04%)
Postgraduate	16 (10.46%)	36 (10.53%)	1 (3.03%)	53 (9.33%)
**Childhood trauma**				
Childhood physical abuse (CPA)	7.03 (3.33)	7.30 (3.70)	8.28 (5.38)	7.28 (3.77)
Childhood emotional abuse (CEA)	10.4 (4.81)	10.8 (5.40)	12.8 (6.11)	10.79 (5.35)
Childhood sexual abuse (CSA)	5.75 (2.32)	6.44 (3.51)	7.81 (4.60)	6.32 (3.37)
Childhood physical neglect (CPN)	7.38 (2.89)	8.21 (3.26)	9.64 (3.83)	8.08 (3.26)
Childhood emotional neglect (CEN)	11.8 (4.92)	11.9 (5.14)	12.4 (4.54)	11.88 (5.07)
**Clinical variables**				
AUDIT score	9.22 (6.75)	28.9 (4.49)	26.6 (9.49)	23.22 (10.43)
DES-taxon belonging	0	0	36 (100%)	36 (6.34%)
BDI score	9.89 (7.44)	11.4 (7.22)	17.2 (7.02)	11.39 (7.46)
STAI-state	41.6 (15.2)	41.7 (14.1)	53.8 (13.6)	42.49 (14.66)
STAI-trait	48.8 (11.4)	51.4 (10.6)	57.6 (11.5)	51.08 (11.11)
PCL-5 Criterion B	6.98 (5.57)	7.68 (5.28)	12.3 (4.81)	7.82 (5.56)
PCL-5 Criterion C	3.15 (2.56)	3.45 (2.30)	4.98 (2.01)	3.48 (2.44)
PCL-5 Criterion D	10.3 (7.40)	11.5 (6.28)	16.3 (5.06)	11.49 (6.77)
PCL-5 Criterion E	8.52 (5.97)	9.67 (5.39)	13.0 (5.30)	9.60 (5.73)

**Note. LALD**: “Low Alcohol Low Dissociation” subgroup; **HALD**: “High Alcohol Low Dissociation” subgroup; **HAHD**: “High Alcohol High Dissociation” subgroup.

**Table 2 brainsci-12-01483-t002:** Childhood trauma and clinical variables by cluster (adjusted models).

Variables	Total Effect	LALD versus HAHD	HALD versus HAHD	LALD versus HALD
	*F* (df)*^p^*^-Value^	Est. (SE)*^p^*^-Value^	Est. (SE)*^p-^*^Value^	Est. (SE)*^p^*^-Value^
**Childhood trauma**				
Physical abuse ^a,b,d^	1.09 (2, 562) ^0.34^	-	-	-
Emotional abuse ^a,b,c^	**3.03 (2, 562) ^<0.05^**	**−2.26 (0.92) ^0.04^**	1.79 (0.87) ^0.09^	0.47 (0.47) ^0.57^
Sexual abuse ^a,b,c,d^	**5.75 (2, 561) ^0.003^**	**−1.86 (0.58) ^0.004^**	−1.21 (0.55) ^0.07^	−0.65 (0.30) ^0.07^
Physical neglect ^b,c^	**8.50 (2, 563) ^<0.001^**	**−2.26 (0.59) ^<0.001^**	**−1.44 (0.55) ^0.03^**	**−0.82 (0.30) ^0.02^**
Emotional neglect ^a,b,c,d^	0.12 (2, 561) ^0.89^	-	-	-

**Clinical variables**				
AUDIT score ^a,b,c,d^	**674.67 (2, 561) ^<0.001^**	**−17.32 (1.04) ^<0.001^**	2.19 (0.98) ^0.07^	**−19.51 (0.53) ^<0.001^**
BDI score ^b,c,d^	**15.70 (2, 562) ^<0.001^**	**−7.29 (1.30) ^<0.001^**	**−5.83 (1.23) ^<0.001^**	−1.45 (0.67) ^0.07^
STAI-state ^a,b,c^	**12.16 (2, 562) ^<0.001^**	**−11.67 (2.57) ^<0.001^**	**−11.88 (2.43) ^<0.001^**	−0.21 (1.32) ^0.99^
STAI-trait ^a,b,c,d^	**11.26 (2, 561) ^<0.001^**	**−9.08 (1.97) ^<0.001^**	**−6.34 (1.86) ^0.002^**	**−2.74 (1.02) ^0.02^**
PCL-5 Criterion B ^a,b^	**14.34 (2, 563) ^<0.001^**	**−5.13 (0.96) ^<0.001^**	**−4.43 (0.91) ^<0.001^**	−0.70 (0.49) ^0.32^
PCL-5 Criterion C ^b,d^	**10.36 (2, 563) ^<0.001^**	**−1.94 (0.43) ^<0.001^**	**−1.55 (0.40) ^<0.001^**	−0.39 (0.22) ^0.16^
PCL-5 Criterion D ^a,b,c^	**12.29 (2, 562) ^<0.001^**	**−5.84 (1.18) ^<0.001^**	**−4.69 (1.11) ^<0.001^**	−1.14 (0.61) ^0.14^
PCL-5 Criterion E ^a,b,c^	**9.04 (2, 562) ^<0.001^**	**−4.16 (0.99) ^<0.001^**	**−3.11 (0.93) ^0.003^**	−1.05 (0.51) ^0.10^

**Note. LALD**: “Low Alcohol Low Dissociation” subgroup; **HALD**: “High Alcohol Low Dissociation” subgroup; **HAHD**: “High Alcohol High Dissociation” subgroup; df: degrees of freedom; Est.: estimates; SE: standard errors; ^a^: adjusted on age; ^b^: adjusted on gender; ^c^: adjusted on marital status; ^d^: adjusted on education; significant results are in bold.

## Data Availability

Data are available on request.

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
