# Peer review of "Childhood Trauma and Dissociation Correlates in Alcohol Use Disorder: A Cross-Sectional Study in a Sample of 587 French Subjects Hospitalized in a Rehabilitation Center"

_brainsci, 2022, doi:10.3390/brainsci12111483_

Round 1
Reviewer 1 Report
Review of manuscript entitled: “Childhood trauma and dissociation correlates in Alcohol Use Disorder: A cross-sectional study in a sample of 587 subjects hospitalized in a rehabilitation center” authored by Grégoire Baudin, Servane Barrault, Hussein El Ayoubi, François Kazour, Nicolas Ballon, Damien Maugé, Coraline Hingray, Paul Brunault and Wissam El Hage
Thank you for possibility to review this interesting manuscript.
Major concerns:
- Did authors take into consideration when childhood trauma occurred (age of patient)?
- Authors stated that “A total of 568 participants (87 women, 15.3%) were included.”, however when we analyze Table 1, we can see that for “Martial status”, total number of participants was not equal to 568, nor in “Highest diploma”. Did authors exclude some participants? What was the criteria of exclusion? If I am missing explanation for this in the manuscript, please forgive me
Minor concerns:
- Title – I would suggest to add that study was performed on French population
- Line 28 – “PTSD” appears for the first time in the text, please provide full explanation of this abbreviation
- Line 70 – “couldn’t”, please try to avoid such abbreviations, should be “could not”
- Table 1 – footnote does not apply to this table I believe, since I cannot see explained “abbreviations” in this table (df, Est., SE, a, b, c, d)
Author Response
Cover Letter
Dear Editor-in-Chief,
Thank you for giving us the opportunity to submit a revised version of our article entitled “Childhood trauma and dissociation correlates in Alcohol Use Disorder: A cross-sectional study in a sample of 587 French subjects hospitalized in a rehabilitation center”.
We are grateful for the comments the reviewers made. We agree with most of them. We feel that the changes they suggested will improve and clarify the presentation of our data.
Please find below the questions raised by the reviewers, and the changes we made in response to their suggestions or, when we decided not to change the initial manuscript, the justification of our decision.
Paul Brunault and Grégoire Baudin, on behalf of the other authors.
University of Paris and University Hospital of Tours, France
REVIEWER 1
Major concerns:
Point 1 : Did authors take into consideration when childhood trauma occurred (age of patient)?
We agree with Reviewer 1 that age of exposure to childhood trauma (CT) is an important aspect to consider. In this study, we used the Childhood Trauma Questionnaire (CTQ) to assess CT. All items of this questionnaire begin with the phrase “When I was growing up ...”. We added this precision in the Methods section. Unfortunately, the CTQ does not allow to know precisely when CT occurred. However, the manual of the CTQ states that “the magnitude of a CTQ scale total score is influenced by several dimensions of maltreatment experiences, including their severity, frequency, and duration”. We addressed this limitation in the Discussion section, by adding the following sentences: “Furthermore, the CTQ does not provide specific informations about the age of exposure to CT. Taking into account such informations would probably lead to a better understanding of the association between CT, alcohol use disorder, dissociation, and psychiatric symptoms.”.
Point 2 : Authors stated that “A total of 568 participants (87 women, 15.3%) were included.”, however when we analyze Table 1, we can see that for “Martial status”, total number of participants was not equal to 568, nor in “Highest diploma”. Did authors exclude some participants? What was the criteria of exclusion? If I am missing explanation for this in the manuscript, please forgive me
Thank you for your vigilance, and for giving us this opportunity to make our manuscript clearer. As noted by Reviewer 1, some of our descriptive statistics are based on fewer than 568 participants. This is due to missing data. In order not to mislead the reader, we reported the missing data rate of our variables of interest in the Results section, by adding the following sentences: “Some of our variables of interest had missing data. Marital status and Highest diploma had 1.58% and 7.04% of missing data respectively. CTQ scores had a rate of missing data ranging from 1.58% (childhood emotional neglect) to 3.87% (childhood emotional abuse). Childhood Physical Abuse, Childhood Sexual Abuse, and Childhood Physical Neglect had missing data rates of 2.29%, 2.82%, and 1.76% respectively. DES-taxon belonging had a missing data rate of 29.40%. Age, Gender, and AUDIT score had no missing data.”. We already acknowledged this limitation in the Discussion section.
Minor concerns:
Point 3 : Title – I would suggest to add that study was performed on French population
Point taken, we changed the title as follows (change appears in bold): Childhood trauma and dissociation correlates in Alcohol Use Disorder: A cross-sectional study in a sample of 587 French subjects hospitalized in a rehabilitation center.
Point 4 : Line 28 – “PTSD” appears for the first time in the text, please provide full explanation of this abbreviation
Point taken, we added “post-traumatic stress disorder” before the first appearance of “PTSD” in the text.
Point 5 : Line 70 – “couldn’t”, please try to avoid such abbreviations, should be “could not”.
Point taken, we made the change accordingly.
Point 6 : Table 1 – footnote does not apply to this table I believe, since I cannot see explained “abbreviations” in this table (df, Est., SE, a, b, c, d)
Thank you for your vigilance, we removed the abovementioned abbreviations in the footnote.
We would like to thank you and the reviewers very much for your work and for taking the time to review and comment our manuscript. We hope that the modifications proposed improved the manuscript’s clarity and quality.
Reviewer 2 Report
I read the paper with many interest and I think it could be interesting for readers after some revisions.
2. Materials and Methods 2.1. Setting and Participants: lt is important to have all completely information about participant in this section (number of participant, age etc.)
Also some questionnaire choice I think not very adequate, in particular Childhood Trauma Questionnaire (CTQ) is more indicate for a younger sample, while you refer that the mean age is 44.5 with SD 9.0. I'm not sure that the reasearch design is completely adequate.
It isn't clear the choise of not use Classical clustering methods, you refer to mixed method but I haven't identified qualitative data in your research. If there are also qualitativ data, please clarify the specif section and data.
Author Response
Cover Letter
Dear Editor-in-Chief,
Thank you for giving us the opportunity to submit a revised version of our article entitled “Childhood trauma and dissociation correlates in Alcohol Use Disorder: A cross-sectional study in a sample of 587 French subjects hospitalized in a rehabilitation center”.
We are grateful for the comments the reviewers made. We agree with most of them. We feel that the changes they suggested will improve and clarify the presentation of our data.
Please find below the questions raised by the reviewers, and the changes we made in response to their suggestions or, when we decided not to change the initial manuscript, the justification of our decision.
Paul Brunault and Grégoire Baudin, on behalf of the other authors.
University of Paris and University Hospital of Tours, France
REVIEWER 2
Point 7: I read the paper with many interest and I think it could be interesting for readers after some revisions.
We would like to thank Reviewer 2 for his/her interest for our manuscript and his/her encouragement.
Point 8: 2. Materials and Methods 2.1. Setting and Participants: lt is important to have all completely information about participant in this section (number of participant, age etc.)
Point taken, we added the following sentence in the Setting and Participants section: “In total, we recruited 568 participants (87 women, 15.3%), with a mean age of 44.5 (SD = 9.0).”.
Point 9: Also some questionnaire choice I think not very adequate, in particular Childhood Trauma Questionnaire (CTQ) is more indicate for a younger sample, while you refer that the mean age is 44.5 with SD 9.0. I'm not sure that the reasearch design is completely adequate.
According to its validation article (Bernstein et al., 2003), the Childhood Trauma Questionnaire – Short Form (CTQ-SF) is a self-administered inventory that retrospectively assesses experiences of childhood maltreatment. The validation was conducted with four samples of participants: one with adult substance abusing patients (mean age = 40.2, sd = 8.8), one with adolescent psychiatric inpatients (mean age = 14.9, sd = 1.4), one with adult substance abusers in the community (mean age = 34, sd not provided), and one normative community sample of adults (mean age = 34.9, sd not provided, range = 33-37 years). For more readability, and as we mentioned in Point 1 (raised by Reviewer 1), we specified in the Measure section that “This 25-item self-rating scale retrospectively assesses five types of childhood maltreatment, namely childhood physical abuse (CPA), emotional abuse (CEA), sexual abuse (CSA), physical neglect (CPN), and emotional neglect (CEN).” (changes appear in bold). We also added the following sentence: “All items begin with the phrase ‘When I was growing up, …’.”
Point 10: It isn't clear the choise of not use Classical clustering methods, you refer to mixed method but I haven't identified qualitative data in your research. If there are also qualitative data, please clarify the specific section and data.
We believe Reviewer 2 raised this point because our wording was not clear enough. To make our message more understandable, we changed two sentences in Statistical Analyses section (changes appear in bold): “DES-taxon belonging was considered a binary variable (i.e. 0: absent, or 1: present), and the total AUDIT score was considered a numerical variable. Classical clustering methods are not suitable for mixed data (i.e., both binary and numerical variables).”
We would like to thank you and the reviewers very much for your work and for taking the time to review and comment our manuscript. We hope that the modifications proposed improved the manuscript’s clarity and quality.